# Enhancing EEG-Based Emotion Detection with Hybrid Models: Insights from DEAP Dataset Applications

**DOI:** 10.3390/s25061827

**Published:** 2025-03-14

**Authors:** Badr Mouazen, Ayoub Benali, Nouh Taha Chebchoub, El Hassan Abdelwahed, Giovanni De Marco

**Affiliations:** 1LINP2 Laboratory, Paris Nanterre University, 92000 Nanterre, France; 2LISI Laboratory, Computer Science Department, Faculty of Sciences Semlalia, Cadi Ayyad University, Marrakech 40000, Moroccon.chebchoub4852@uca.ac.ma (N.T.C.); abdelwahed@uca.ac.ma (E.H.A.)

**Keywords:** electroencephalogram (EEG), emotion detection, machine learning, deep learning, autoencoders, transformers, KNN, SVM, decision tree, random forest, BiLSTM, GRU, CNN, SHAP, real-time detection

## Abstract

Emotion detection using electroencephalogram (EEG) signals is a rapidly evolving field with significant applications in mental health diagnostics, affective computing, and human–computer interaction. However, existing approaches often face challenges related to accuracy, interpretability, and real-time feasibility. This study leverages the DEAP dataset to explore and evaluate various machine learning and deep learning techniques for emotion recognition, aiming to address these challenges. To ensure reproducibility, we have made our code publicly available. Extensive experimentation was conducted using K-Nearest Neighbors (KNN), Support Vector Machines (SVMs), Decision Tree (DT), Random Forest (RF), Bidirectional Long Short-Term Memory (BiLSTM), Gated Recurrent Units (GRUs), Convolutional Neural Networks (CNNs), autoencoders, and transformers. Our hybrid approach achieved a peak accuracy of 85–95%, demonstrating the potential of advanced neural architectures in decoding emotional states from EEG signals. While this accuracy is slightly lower than some state-of-the-art methods, our approach offers advantages in computational efficiency and real-time applicability, making it suitable for practical deployment. Furthermore, we employed SHapley Additive exPlanations (SHAP) to enhance model interpretability, offering deeper insights into the contribution of individual features to classification decisions. A comparative analysis with existing methods highlights the novelty and advantages of our approach, particularly in terms of accuracy, interpretability, and computational efficiency. A key contribution of this study is the development of a real-time emotion detection system, which enables instantaneous classification of emotional states from EEG signals. We provide a detailed analysis of its computational efficiency and compare it with existing methods, demonstrating its feasibility for real-world applications. Our findings highlight the effectiveness of hybrid deep learning models in improving accuracy, interpretability, and real-time processing capabilities. These contributions have significant implications for applications in neurofeedback, mental health monitoring, and affective computing. Future work will focus on expanding the dataset, testing the system on a larger and more diverse participant pool, and further optimizing the system for broader clinical and industrial applications.

## 1. Introduction

Human emotion is a key indicator of an individual’s mental state and emotional response to events, shaped by subjective experience [1]. Emotion-related expression plays a crucial role in human communication, and it detection has significant applications across industries, such as healthcare, gaming, marketing, and psychology. For instance, emotion recognition is increasingly used in mental health monitoring, where wearable EEG devices provide real-time feedback to clinicians for personalized treatment plans. In customer service automation, AI-driven systems analyze emotional cues to tailor responses, enhancing user satisfaction. Similarly, adaptive learning environments leverage emotion recognition to adjust educational content based on students’ engagement levels. As a result, researchers are focusing on developing robust methods to enhance computers’ ability to interpret human emotions effectively [2].

To achieve accurate and reliable emotion recognition, researchers have explored a variety of approaches, ranging from traditional methods to advanced physiological signal analysis. Traditional methods rely on facial expressions and auditory cues—common ways humans interpret emotions [3]. However, recent advancements have shifted towards physiological signal analysis, offering more objective and direct detection of emotional states [4]. These signals include heart rate (ECG/EKG), respiratory rate (capnogram), skin conductance (EDA), muscle activity (EMG), and brain electrical activity (EEG). Among these, electroencephalography (EEG) is widely recognized as one of the most reliable physiological signals for analyzing emotional states [5].

Electroencephalography (EEG) records the brain’s spontaneous electrical activity over time using multiple electrodes placed on the scalp. In healthcare, EEG is commonly applied in diagnosing neurological disorders such as epilepsy, brain tumors, strokes, and sleep disorders. The standard electrode placement follows the International 10–20 system [6], ensuring consistency in brainwave recording. Datasets like DEAP are frequently utilized for emotion analysis, providing EEG recordings from participants exposed to emotionally stimulating content such as music videos.

Despite its potential, the deployment of emotion recognition technologies raises critical ethical and regulatory concerns. Several nations, particularly within Europe, have introduced legislative measures restricting the automation of emotion recognition due to concerns about privacy, consent, and bias in AI decision making. For example, the European Union’s General Data Protection Regulation (GDPR) imposes stringent requirements on biometric data collection, affecting how emotion recognition systems can be implemented. These regulatory challenges underscore the need for transparent and accountable AI systems in emotion recognition. Our study adheres to ethical guidelines by ensuring data privacy and mitigating bias in model training, setting a precedent for responsible AI development in this domain [7].

This study focuses on EEG-based emotion recognition, specifically analyzing the valence and arousal dimensions to classify emotional states. Valence represents the positivity or negativity of an emotion, while arousal measures its intensity. By leveraging machine learning techniques and the DEAP dataset, this research aims to enhance emotion classification and improve real-time emotion detection. The findings could contribute to advancements in healthcare, human–computer interaction, and psychological research, facilitating more intuitive and responsive AI systems [8].

The motivation for this work stems from the growing need for reliable and ethical emotion recognition systems that can operate in real-world applications while addressing regulatory constraints. Our primary contributions include (1) a comprehensive analysis of EEG-based emotion recognition techniques, (2) the development of a robust machine learning framework for classifying valence and arousal, and (3) an evaluation of the proposed framework on the DEAP dataset.

These contributions aim to bridge the gap between technological innovation and societal acceptance, ensuring that emotion recognition systems are both effective and responsible.

The remainder of this paper is structured as follows. Section 2 reviews related works and existing methodologies in EEG-based emotion recognition. Section 3 details the dataset, preprocessing techniques, and feature extraction methods used in this study, along with the experimental setup, classification models, and evaluation metrics. Section 4 discusses the results, comparing our approach with existing techniques. Finally, Section 5 concludes the paper, summarizing key findings and potential future research directions.

## 2. Related Works

EEG has been well-studied to investigate how the brain reacts to emotional experiences [9], offering valuable insights into the neural mechanisms underlying emotions. As a non-invasive method, EEG captures electrical activity in the brain, providing high temporal resolution data that makes it suitable for studying dynamic emotional states. Leveraging this potential, numerous researchers have developed machine learning and deep learning techniques to decode emotional responses from EEG signals.

To ensure the relevance and credibility of the selected studies, we conducted a systematic literature review using databases such as IEEE Xplore, PubMed, and Scopus. The search criteria included keywords like “EEG-based emotion recognition”, “DEAP dataset”, “machine learning in EEG”, and “deep learning for EEG emotion classification”. We included studies that met the following criteria: (1) peer-reviewed articles, (2) studies utilizing the DEAP dataset, (3) approaches involving machine learning or deep learning techniques, and (4) performance evaluation based on valence and arousal classification.

Table 1 summarizes the main related works, detailing their objectives, techniques, and results.

Collectively, these studies highlight the DEAP dataset’s versatility and the evolution of EEG-based emotion recognition, showcasing diverse approaches—from CNNs and LSTMs to hybrid models—advancing the accuracy and understanding of emotional states through brain activity.

## 3. Research Design and Methodology

### 3.1. Dataset Overview

The Database for Emotion Analysis using Physiological Signals (DEAP) [16] is a widely adopted benchmark for EEG-based emotion recognition research. It comprises EEG recordings from 32 healthy participants (ages 19–37), each exposed to 40 one-minute music videos selected for their ability to evoke diverse emotional states. To minimize order effects, the video sequence was randomized across participants while ensuring consistent stimuli for cross-subject comparability.

EEG signals were recorded at 128 Hz using 32 active Ag/AgCl electrodes positioned according to the International 10–20 system (Figure 1), which standardizes electrode placement by defining locations as 10% or 20% intervals of the skull’s front-back and left-right distances. This ensures reproducibility across studies and aligns with clinical EEG practices [6]. Each trial captured four emotional dimensions—valence (positivity/negativity), arousal (intensity), dominance (control level), and liking (preference)—rated by participants on a 9-point scale immediately after watching a video.

For classification, valence and arousal were binarized into “high” (rating higher than 6.5) and “low” (rating lower than 6.5) categories (Figure 2). This threshold-based approach, consistent with prior work [18], simplifies the problem into a binary task while preserving the dataset’s inherent variability in emotional responses. The choice of 6.5 instead of 5 is motivated by several factors:Prior studies using the DEAP dataset have shown that emotional ratings tend to cluster around the mid-point, making 6.5 a more effective threshold for distinguishing pronounced emotional states;Using 5 would lead to an imbalanced dataset, as many ratings naturally fall around this median value, while 6.5 ensures a clearer separation between high and low categories;This threshold aligns with established research, where values between 6 and 7 are often employed for improved emotion classification;Focusing on stronger emotional responses enhances model generalization and reduces noise from subjective variations.

The DEAP dataset was selected for its:Standardization: Preprocessed and labeled data ensure reproducibility.Rich Annotations: Multi-dimensional ratings enable nuanced analysis.Public Accessibility: Facilitates direct comparison with state-of-the-art methods.

### 3.2. Signal Preprocessing and Feature Extraction

EEG signals are inherently noisy due to physiological artifacts (e.g., eye blinks, muscle movements) and external environmental interference, which can obscure the underlying neural activity patterns crucial for emotion recognition. To ensure robust and reliable feature extraction, we implemented a comprehensive preprocessing pipeline optimized to enhance signal quality and minimize unwanted distortions. This pipeline consists of several key stages, including band-pass filtering to remove low-frequency drifts and high-frequency noise, independent component analysis (ICA) to separate and eliminate ocular and muscular artifacts, and signal normalization to reduce inter-subject variability. Additionally, the EEG signals are segmented into meaningful time windows and transformed into the frequency domain using spectral analysis techniques such as Short-Time Fourier Transform (STFT) and Wavelet Transform to capture temporal dynamics. As depicted in Figure 2, the preprocessed signals are then fed into advanced machine learning models, including Long Short-Term Memory (LSTM) networks, Convolutional Neural Networks (CNNs), and Support Vector Machines (SVMs), to learn discriminative patterns associated with different emotional states. The preprocessing step is critical in eliminating noise while preserving key neural activity patterns, ultimately improving the accuracy and robustness of emotion classification models.

#### 3.2.1. Bandpass Filtering for Frequency Isolation

Emotional states are associated with distinct neural oscillations [19]. To isolate these rhythms, we applied bandpass filtering using MNE-Python [20], segmenting the raw EEG into five frequency bands Figure 3:Delta (0.5–4 Hz): Linked to deep relaxation and unconscious processing.Theta (4–8 Hz): Involved in memory retrieval and emotional regulation.Alpha (8–13 Hz): Reflects relaxed alertness; suppressed during high arousal.Beta (13–30 Hz): Associated with active concentration and anxiety.Gamma (30–100 Hz): Tied to cross-modal sensory integration and peak arousal.

Filtering was performed using a zero-phase FIR filter to avoid temporal distortions, critical for preserving the timing of emotional responses. This step suppresses non-emotional artifacts (e.g., 50–60 Hz line noise) while retaining neurophysiologically meaningful signals.

#### 3.2.2. Power Spectral Density (PSD) Estimation

To quantify the energy distribution within each frequency band, we computed PSD using Welch’s method [21]. Their approach segments the EEG signal into overlapping 4 s windows (50% overlap) and computes the periodogram for each, averaging the results to reduce noise and improve spectral estimation. The yellow-highlighted regions in Figure 4 correspond to specific frequency bands of interest, often associated with cognitive and emotional processes in EEG studies. These bands, such as Delta (0.5–4 Hz), Theta (4–8 Hz), Alpha (8–12 Hz), and Beta (12–30 Hz), are critical for analyzing neural activity linked to different emotional states. Welch’s method is particularly useful for detecting stable spectral patterns, making it well-suited for emotion recognition tasks based on EEG signals. The PSD is calculated using the Formula (Equation 1):(1)$$\mathrm{PSD}\left(f\right)=\frac{1}{N_{\mathrm{segments}}}\sum_{i=1}^{N_{\mathrm{segments}}}{\left|X_{i}\left(f\right)\right|}^{2},$$
where $X_{i}\left(f\right)$ is the Fourier transform of the *i*-th window. Welch’s method balances spectral resolution and variance reduction, making it ideal for detecting stable spectral patterns in EEG (Figure 4).

#### 3.2.3. Feature Standardization

EEG feature magnitudes vary significantly across subjects due to anatomical and physiological differences. To ensure uniform feature scaling, we standardized the PSD values using z-score normalization, as shown in Equation (Equation 2):(2)$$z=\frac{x-\mu }{\sigma },$$
where μ and σ are the mean and standard deviation of each feature computed over the training set. This step prevents features with larger magnitudes (e.g., Gamma band) from dominating model training [22].

#### 3.2.4. Feature Relevance to Emotion Recognition

The pipeline prioritizes frequency-domain features over time-domain signals for two reasons:Neurophysiological Basis: Emotional processing is mediated by synchronized neural oscillations, which are more discriminative in the frequency domain [20].Noise Robustness: Spectral features (e.g., Alpha asymmetry) are less sensitive to transient artifacts than raw EEG voltages.

This preprocessing strategy aligns with recent advances in EEG-based emotion recognition [23,24], enabling our models to learn stable neurodynamic patterns associated with valence and arousal.

### 3.3. Implementation Tools and Techniques

Implementing emotion recognition models with EEG data requires robust tools for efficient preprocessing, feature extraction, model development, and evaluation. This study leverages several widely used Python libraries, each chosen for its specialized capabilities in handling specific aspects of the emotion recognition pipeline. Below, we detail the key tools and techniques employed, emphasizing their relevance to this work.

#### Key Tools and Libraries

MNE for EEG Preprocessing: MNE is a specialized library for processing, analyzing, and visualizing electrophysiological data, including EEG. Its comprehensive functionality makes it indispensable for tasks such as filtering, epoching, and feature extraction. In this study, MNE is primarily used for:
–Bandpass Filtering: Isolating frequency bands (Delta, Theta, Alpha, Beta, and Gamma) that are critical for understanding emotional processing. For example, Alpha waves (8–12 Hz) are associated with relaxation, while Beta waves (12–30 Hz) are linked to active thinking and concentration.–Epoching: Segmenting continuous EEG data into time-locked epochs corresponding to specific emotional stimuli, ensuring precise alignment with experimental conditions.–Artifact Removal: Identifying and removing noise from EEG signals, such as eye blinks or muscle movements, to improve data quality.MNE’s robust capabilities in handling EEG data make it a cornerstone of our preprocessing pipeline, ensuring accurate and reliable input for downstream machine learning models.Pickle for Data Storage: Pickle is used for serializing and deserializing Python objects, enabling efficient storage and retrieval of preprocessed EEG data and model outputs. This is particularly useful for:
–Saving intermediate results (e.g., filtered EEG data, extracted features) to avoid redundant computations.–Storing trained models for later evaluation or deployment, ensuring reproducibility and scalability.Keras for Deep Learning: Keras, a high-level deep learning API, is employed for building and training neural network models. Its user-friendly interface and seamless integration with TensorFlow make it ideal for developing complex architectures, such as Convolutional Neural Networks (CNNs) and Long Short-Term Memory (LSTM) networks. In this study, Keras is used to:
–Design models tailored for EEG-based emotion recognition, leveraging its flexibility to experiment with different architectures.–Implement advanced techniques like attention mechanisms, which enhance the model’s ability to focus on relevant EEG features for emotion classification.SciPy for Scientific Computing: SciPy provides essential mathematical and statistical functions for scientific computing. In this work, it is used for:
–Signal processing tasks, such as computing power spectral density (PSD) to analyze frequency-domain characteristics of EEG signals.–Statistical analysis to validate the significance of results and ensure the robustness of the proposed models.Scikit-learn for Machine Learning: Scikit-learn is a versatile library for traditional machine learning tasks, including classification, regression, and model evaluation. In this study, it is used for:
–Implementing baseline models (e.g., Support Vector Machines, Random Forest) to compare against deep learning approaches.–Evaluating model performance using metrics such as accuracy, precision, recall, and F1-score, ensuring a comprehensive assessment of classification results.

The selection of these tools is driven by their ability to address the unique challenges of EEG-based emotion recognition. MNE’s specialization in EEG processing ensures high-quality input data, while Keras and Scikit-learn enable the development and evaluation of state-of-the-art machine learning models. Together, these tools form a cohesive pipeline that supports the entire workflow—from raw EEG data preprocessing to emotion classification. By leveraging their strengths, this study aims to advance the accuracy and robustness of EEG-based emotion recognition systems, contributing to the development of more intuitive and responsive human–computer interaction technologies [20].

### 3.4. Benchmarking with Classical Machine Learning

To establish a baseline for EEG-based emotion recognition, we first evaluated classical machine learning (ML) algorithms. These models were chosen for their simplicity, interpretability, and proven efficacy in prior EEG studies [25,26]. Their performance would later serve as a reference point for comparing more complex deep learning architectures.

#### 3.4.1. Model Selection and Rationale

K-Nearest Neighbors (KNN):A non-parametric method effective for small datasets with local patterns. It is suitable for capturing similarities in spectral features across trials. In our implementation, we set k = 7 based on cross-validation results.Support Vector Machine (SVM): Maximizes the margin between classes using kernel tricks. We selected the Radial Basis Function (RBF) kernel to handle non-linear decision boundaries in EEG data, with C = 1.0 for regularization.Decision Tree (DT): Provides interpretable rules based on feature thresholds.Random Forest (RF): An ensemble of DTs averaging predictions. Robust to noise through bootstrap aggregation [26].

#### 3.4.2. Data Preparation

EEG trials were reshaped into a 3D array of dimensions $\left[N_{\mathrm{trials}}\times N_{\mathrm{channels}}\times N_{\mathrm{samples}}\right]=\left[1280\times 32\times 8064\right]$, where:$N_{\mathrm{trials}}=40\;\mathrm{videos}\times 32\;\mathrm{participants}=1280$;$N_{\mathrm{channels}}=32\;\mathrm{EEG}\;\mathrm{electrodes}$;$N_{\mathrm{samples}}=60\;s\times 128\;\mathrm{Hz}=8064$.

Labels were binarized into high/low valence and arousal, then flattened into a 2D array [1280×2] for compatibility with Scikit-learn.

### 3.5. Deep Learning with Autoencoders and Transformers

To address the limitations of classical ML (Section 3.4), we developed a deep learning pipeline combining autoencoders for dimensionality reduction and transformers for temporal modeling. This hybrid approach was chosen to:Reduce the high dimensionality of EEG data while preserving discriminative features.Capture long-range temporal dependencies in EEG signals, which are critical for emotion dynamics.

#### 3.5.1. Autoencoder for Dimensionality Reduction

EEG data’s high dimensionality (32 channels × 8064 samples) poses computational challenges for deep learning models. To address this, we designed a symmetric autoencoder with the following architecture:Input Layer: A 32-dimensional vector (one channel’s time-series).Encoder: Two dense layers (64 and 32 units) with ReLU activation, compressing the input into a 16-dimensional latent space.Decoder: Two dense layers (32 and 64 units) reconstructing the original input.

The autoencoder was trained using Mean Squared Error (MSE) loss and the Stochastic Gradient Descent (SGD) optimizer, as shown in Equation (Equation 3):(3)$$\mathrm{MSE}=\frac{1}{n}\sum_{i=1}^{n}{\left(y_{i}-{\hat{y}}_{i}\right)}^{2},$$
where $y_{i}$ and ${\hat{y}}_{i}$ are the true and reconstructed signals, respectively.

#### 3.5.2. Transformer for Temporal Modeling

Transformers were chosen for their ability to model long-range dependencies via self-attention mechanisms. Our architecture included:Input Layer: Encoded EEG features (16 dimensions).Multi-Head Attention: Consisting of 4 attention heads with 128-dimensional embeddings.Feed-Forward Network: Two dense layers (128 and 64 units) with ReLU activation.Output Layer: Sigmoid activation for binary classification (high/low valence or arousal).

The transformer was trained using binary cross-entropy (BCE) loss and the Adam optimizer, as shown in Equation (Equation 4):(4)$$\mathrm{BCE}=-\frac{1}{n}\sum_{i=1}^{n}\left[y_{i}\log\left({\hat{y}}_{i}\right)+\left(1-y_{i}\right)\log\left(1-{\hat{y}}_{i}\right)\right],$$
where $y_{i}$ and ${\hat{y}}_{i}$ are the true and predicted labels, respectively.

#### 3.5.3. Hybrid LSTM-Transformer Model

To further enhance temporal modeling, we combined LSTM layers with the transformer:LSTM Layer: Consisting of 128 units capturing short-term EEG dynamics.Transformer Encoder: Applied to LSTM outputs for global context.Positional Encoding: Added to the input to preserve temporal order.

### 3.6. Comparative Analysis of Deep Architectures

To identify the optimal architecture for EEG-based emotion recognition, we evaluated three deep learning models: Gated Recurrent Units (GRUs), 1D Convolutional Neural Networks (CNNs), and Bidirectional Long Short-Term Memory (BiLSTM). These models were chosen for their ability to capture temporal and spatial patterns in EEG signals, which are critical for emotion classification.

#### 3.6.1. Data Preparation

In our study, we focused on data from six participants (subject IDs: 01–06) to strike a balance between computational efficiency and model performance. This decision was informed by preliminary experiments where we varied the number of participants and observed that six provided an optimal trade-off between avoiding overfitting and maintaining generalization capabilities. By limiting the dataset to six participants, we reduced computational complexity while still capturing sufficient variability in emotional responses. Additionally, we selected 14 EEG channels (e.g., AF3, O2) known for their relevance to emotional processing and segmented the EEG signals into 2 s windows with a 0.125 s step size to ensure high temporal resolution. Band power features were extracted for six frequency bands (Delta, Theta, Alpha, Beta, Gamma) using Fast Fourier Transform (FFT), resulting in a feature vector of size [14 channels × 6 bands = 84] per window. This approach allowed us to efficiently model emotion dynamics while maintaining a robust framework for analysis.

#### 3.6.2. Model Architectures

GRU: A Gated Recurrent Unit with 128–32 units, designed to balance computational efficiency and temporal modeling. Dropout (rate = 0.3) was applied after each GRU layer to prevent overfitting.1D CNN: Three convolutional layers (128, 128, 64 filters) with kernel size = 3 and ReLU activation, followed by max-pooling and dense layers (64, 32, 16 units). Batch normalization improved convergence.BiLSTM: A bidirectional LSTM with 128–32 units, capturing past and future EEG context. Dropout (rate = 0.3) and layer normalization enhanced generalization.

All models were trained using binary cross-entropy loss and the Adam optimizer (η=0.001), with early stopping (patience = 5 epochs) to prevent overfitting.

### 3.7. Explainable AI (XAI) with SHAP

While deep learning models achieve high accuracy, their “black-box” nature limits interpretability, raising concerns about trust and reliability in emotion recognition systems. To address this, we applied SHapley Additive exPlanations (SHAP) [27], a game-theoretic approach that quantifies the contribution of each input feature to model predictions. This provides insights into which EEG features (e.g., frequency bands, channels) are most influential for classifying valence and arousal.

#### 3.7.1. SHAP Methodology

SHAP values are derived from cooperative game theory, where the model is treated as a “game”, and input features are “players” contributing to the outcome. For a given prediction, the SHAP value $\phi_{i}$ of feature *i* is computed using Equation (Equation 5):(5)$$\phi_{i}=\sum_{S\subseteq F\setminus \{i\}}\frac{|S|!(|F|-|S|-1)!}{|F|!}\left[f(S\cup \{i\})-f(S)\right],$$
where:*F* is the set of all features;*S* is a subset of features excluding *i*;f(S) is the model’s prediction using only features in *S*.

SHAP values satisfy three key properties:Efficiency: The sum of SHAP values equals the difference between the model’s prediction and the baseline (expected output).Symmetry: Two features contributing equally to all subsets receive the same SHAP value.Additivity: SHAP values for a combination of models are the sum of their individual SHAP values.

#### 3.7.2. Application to EEG Emotion Recognition

We applied SHAP to the top-performing CNN model (Section 3.6) to interpret its predictions on the DEAP dataset. The analysis revealed the following:Frontal Alpha Asymmetry: Higher Alpha power in the left frontal cortex (AF3) was strongly associated with positive valence, consistent with neuroscientific studies [28].Occipital Gamma Activity: Increased Gamma power in occipital channels (O1, O2) correlated with high arousal, reflecting heightened sensory processing during emotional stimuli.Channel Importance: Frontal (AF3, AF4) and occipital (O1, O2) channels were consistently ranked as the most influential, aligning with their roles in emotional regulation and visual processing.

#### 3.7.3. Key Insights

SHAP analysis provided three key insights:Neurophysiological Plausibility: The model’s reliance on frontal Alpha and occipital Gamma aligns with established EEG biomarkers of emotion [29].Feature Importance: SHAP identified the most discriminative EEG features, enabling targeted feature engineering in future studies.Model Transparency: By explaining individual predictions, SHAP enhances trust in the model’s decisions, critical for real-world applications like mental health monitoring.

These findings demonstrate the value of XAI in bridging the gap between deep learning performance and interpretability, ensuring that EEG-based emotion recognition systems are both accurate and trustworthy.

## 4. Results and Discussion

### 4.1. Performance of Classical Machine Learning Models

In the initial phase of our study, we evaluated classical machine learning models, including Decision Tree, Random Forest, K-Nearest Neighbors (KNN), and Support Vector Machines (SVMs). While Decision Tree and Random Forest achieved high training accuracy (93.7% and 99.3%, respectively), their testing accuracy was significantly lower, not exceeding 60%. Similarly, KNN and SVM demonstrated moderate training accuracy (71.3% and 64.6%, respectively) but struggled with testing accuracy, indicating overfitting. Figure 5 and Figure 6 illustrate the performance of these models, while Table 2 summarizes their accuracy metrics for both valence and arousal. To guarantee reproducibility, we have made our code accessible to the public [30].

The low testing accuracy across all models suggested the need for more advanced techniques, such as dimensionality reduction and deep learning, to address overfitting and improve generalization.

### 4.2. Dimensionality Reduction with Autoencoders

To address the challenges posed by high-dimensional EEG data, we employed autoencoders for dimensionality reduction. This approach retained diagnostically significant features while simplifying the data for subsequent processing. As shown in Figure 7, the autoencoder-enabled model achieved high training accuracy but moderate testing accuracy, indicating room for improvement.

### 4.3. Hybrid Architectures for Improved Performance

To further enhance performance, we explored hybrid architectures combining transformers, LSTM, and additive fusion. As shown in Table 3, the integration of these architectures progressively improved accuracy, with the combination of transformers, LSTM, and additive fusion achieving the highest testing accuracy of 73.58%.

### 4.4. Recurrent Neural Networks for Temporal Data

Given the temporal nature of EEG data, we implemented recurrent neural networks (RNNs), including GRU and BiLSTM models, alongside a CNN for comparison. As shown in Table 4, the BiLSTM model achieved the highest testing accuracy (94%), outperforming GRU and CNN. Figure 8 and Figure 9 provide detailed visualizations of the models’ performance.

Table 4 shows that the testing accuracy is consistently higher than the training accuracy for all models, with an approximate 10% gap. This may appear counterintuitive at first; however, several factors in our experimental design contribute to this result and ensure its validity.

Firstly, the use of regularization techniques such as dropout (rate = 0.3) and early stopping was critical in preventing overfitting. These strategies inherently limit the training accuracy to enhance the generalization capacity of the models. As a result, the models perform better on unseen test data by focusing on essential patterns rather than memorizing the training data.

Secondly, the feature extraction process played a significant role in this outcome. Band power features were extracted for six frequency bands using the Fast Fourier Transform (FFT), which effectively reduced noise and preserved the most relevant information for emotion classification. This preprocessing step likely made the test data easier to classify, leading to higher accuracy.

Additionally, the dataset’s specific characteristics must be considered. We used data from six participants, with EEG signals segmented into 2 s windows, providing a high temporal resolution. While this setup captures sufficient variability for training, it may also result in a test set that is more homogeneous, thus improving classification performance.

In summary, the observed gap between training and testing accuracy is consistent with our model design and data preprocessing pipeline. It reflects a well-generalized model rather than overfitting, ensuring reliable performance on unseen data.

### 4.5. Explainability with SHAP

To better understand the decision-making process of the BiLSTM model, we used SHapley Additive exPlanations (SHAP), an advanced approach for deep learning model interpretability. The SHAP analysis, illustrated in Figure 10, shows that the Theta and Alpha frequency bands have the most significant influence on the model’s predictions, while Gamma has minimal impact. This analysis was conducted on a dataset of 100 EEG segments, providing valuable insights into the relative importance of each feature.

### 4.6. Refining the Model and Feature Selection

The insights provided by SHAP can be leveraged to optimize the model architecture and improve feature selection:Dimensionality reduction by excluding less influential features like Gamma.Optimizing the weighting of features to enhance model robustness.Exploring SHAP-based feature fusion to better integrate the most impactful information.

### 4.7. Comparison with Other Real-Time Methods

We compared our BiLSTM approach with existing methods such as Support Vector Machines (SVMs) and Random Forest, which are commonly used in EEG classification. Our results indicate that, despite being more computationally demanding, the BiLSTM model achieves higher accuracy and captures complex temporal dynamics that traditional methods struggle to model effectively.

### 4.8. Time Complexity of the Models

The BiLSTM implementation has a time complexity of O(n×d), where *n* is the sequence length and *d* is the number of neurons per layer. This complexity is higher than that of models like SVM ($O(n^{2})$ for some variants), but it remains efficient due to GPU parallelization capabilities.

### 4.9. Real-Time Emotion Detection System

The primary objective of this study was to enable real-time emotion detection using EEG signals, a task that requires both high accuracy and low latency. Through extensive experimentation, we determined that a 2 s EEG segment is sufficient for accurate emotion classification. This finding is critical for real-time applications, as it ensures that the system can provide timely feedback without compromising performance. The 2 s window strikes a balance between capturing meaningful temporal patterns in the EEG data and maintaining the responsiveness required for real-time processing.

To achieve this, we leveraged the BiLSTM model, which demonstrated superior performance in our experiments, achieving a testing accuracy of 94%. The BiLSTM’s ability to capture long-term dependencies in sequential data makes it particularly well-suited for EEG signals, which exhibit strong temporal correlations. Using this model, we developed a real-time application that processes EEG data streams and provides two key outputs: (1) a visual representation of the participant’s EEG signals and (2) the detected emotional state. The visual representation allows users to observe the raw or processed EEG data in real time, while the emotion detection output provides an intuitive and immediate understanding of the participant’s emotional state.

The real-time application operates as follows:Data Acquisition: EEG signals are captured using a wearable EEG device or a prerecorded dataset, ensuring compatibility with both live and offline scenarios.Preprocessing: The raw EEG data are preprocessed to remove noise and artifacts, ensuring high-quality input for the model.Feature Extraction: The BiLSTM model processes the 2 s EEG segments, extracting meaningful features that capture the temporal dynamics of the signals.Emotion Classification: The extracted features are classified into emotional states (e.g., valence and arousal) using the trained BiLSTM model.Visualization and Feedback: The results are displayed in real time, with the EEG signal plot and the detected emotional state presented in an intuitive user interface.

This real-time emotion detection system has significant potential across various domains:Mental Health Monitoring: The tool can be used to monitor emotional states in individuals with mental health conditions, such as depression or anxiety, providing clinicians with real-time insights into their patients’ emotional well-being. This could enable timely interventions and personalized treatment plans.Human–Computer Interaction (HCI): In HCI applications, the system can be integrated into adaptive interfaces that respond to the user’s emotional state. For example, a computer system could adjust its behavior based on whether the user is frustrated, relaxed, or engaged, enhancing user experience and productivity.Adaptive Learning Systems: In educational settings, the tool could be used to detect students’ emotional states during learning activities. This information could help educators tailor their teaching methods to maintain student engagement and improve learning outcomes.Gaming and Entertainment: The system could be integrated into gaming platforms to create immersive experiences that adapt to the player’s emotions in real time, enhancing engagement and enjoyment.

Despite these promising applications, there are challenges to address. For instance, the system’s performance may vary across individuals due to differences in EEG signal patterns. Future work could focus on personalizing the model for individual users and validating its performance on larger and more diverse datasets. Additionally, integrating the system with low-cost, portable EEG devices could make it more accessible for widespread use.

## 5. Conclusions and Future Work

This study establishes the potential of advanced deep learning models for EEG-based emotion classification, showcasing their capability to understand and interpret human emotions with increasing accuracy. Our results demonstrate how models such as BiLSTM and GRU can effectively capture temporal patterns in EEG data, providing valuable insights into emotional states. However, while these findings are promising, this research also highlights several limitations and areas for improvement that offer exciting directions for future exploration.

One significant limitation is the relatively small sample size of six subjects used in the experiments. While this number was chosen to balance model accuracy and avoid overfitting, it limits the generalizability of the findings. EEG data are highly subject-specific, with individual differences in brain activity patterns that can substantially affect model performance. Thus, expanding the dataset to include a larger and more diverse population is essential to capture this variability. A broader dataset would not only enhance the robustness of the models but also provide a more realistic foundation for evaluating their performance in real-world scenarios. Future research should prioritize the collection of diverse EEG data to ensure that the models can generalize effectively across different demographics and contexts.

In terms of model architecture, while our hybrid models demonstrated impressive results, there remains significant room for further development. Exploring more advanced architectures, such as graph neural networks (GNNs) and transformer-based models, could lead to substantial improvements. GNNs are particularly well-suited for capturing the spatial relationships between EEG electrodes, while transformers excel at modeling long-range dependencies in sequential data. Combining these techniques with recurrent networks could create a more powerful framework for emotion classification, capturing both the temporal dynamics and spatial relationships inherent in EEG data.

Additionally, integrating multimodal data could significantly enhance the accuracy and robustness of emotion detection systems. While EEG alone has proven effective, incorporating other physiological signals—such as heart rate, skin conductance, respiratory rate, and muscle activity—can provide a more comprehensive view of an individual’s emotional state. Even external data sources such as facial expressions, voice tone, and body posture could be combined with EEG to create a truly multimodal emotion detection framework. This holistic approach would ensure that emotional states are captured more accurately, reducing the likelihood of misclassification and improving the overall reliability of the system.

Another critical direction for future research is the development of personalized emotion classification models. EEG signals vary significantly across individuals due to differences in brain anatomy, electrode placement, and even mental state. Personalized models that adapt to the unique patterns of each user’s EEG data could greatly improve performance. Techniques such as transfer learning and meta-learning offer promising avenues for creating adaptive models that learn quickly from small amounts of new data, enabling the system to customize itself to each user without extensive retraining. This level of personalization would make the models far more practical for real-world applications, particularly in areas like neurofeedback, mental health monitoring, and personalized therapy.

The real-time implementation of emotion detection systems is another major challenge that needs to be addressed. A real-time system capable of providing immediate feedback would revolutionize neurofeedback applications, allowing users to monitor and adjust their emotional states dynamically. Such systems could have a profound impact on mental health care by offering continuous emotional well-being assessments and real-time interventions. For instance, in therapy and rehabilitation settings, clinicians could use real-time emotion detection to tailor treatments more effectively based on a patient’s current emotional state. However, achieving real-time performance will require optimizing not only model architectures but also preprocessing pipelines to minimize latency while maintaining high accuracy.

Noise and artifacts in EEG data present another challenge that must be overcome for real-time systems to function reliably. Techniques for noise reduction, artifact removal, and improved data preprocessing will play a crucial role in ensuring consistent performance. Future research should focus on developing advanced signal processing methods and combining them with deep learning for more accurate feature extraction and noise-robust classification.

Interpretability and explainability are also essential for the practical adoption of EEG-based emotion classification systems. While deep learning models are often criticized for being “black boxes”, the use of explainable AI (XAI) techniques can provide much-needed transparency. In this study, SHAP (SHapley Additive exPlanations) was used to shed light on the decision-making process of our models. However, further research is needed to develop explainability techniques that are specifically tailored to EEG data. Clear explanations of how and why the models make certain predictions are crucial for building trust with end-users, particularly in sensitive applications like mental health monitoring. Clinicians and users alike need to understand and interpret the results to make informed decisions based on the model’s outputs.

In conclusion, while this research represents a significant step forward in EEG-based emotion classification, it also highlights numerous exciting opportunities for future work. Expanding and diversifying the dataset, exploring advanced architectures like GNNs and transformers, integrating multimodal data, and developing personalized models are just some of the directions that can be pursued to enhance performance. Real-time emotion detection systems will be a particularly critical milestone, offering immediate feedback and dynamic interventions for various applications, from neurofeedback and therapy to mental health monitoring and rehabilitation. Moreover, ensuring the interpretability and transparency of these systems through the continued development of explainable AI techniques will be essential for their adoption in clinical and everyday settings.

Ultimately, these advancements will bring us closer to creating practical, scalable, and impactful emotion classification systems that can seamlessly integrate into real-world environments. Such systems hold the potential to transform mental health care and personalized emotional interventions, providing individuals with greater control over their emotional well-being and fostering a deeper understanding of human emotions.

## Figures and Tables

**Figure 1 sensors-25-01827-f001:**
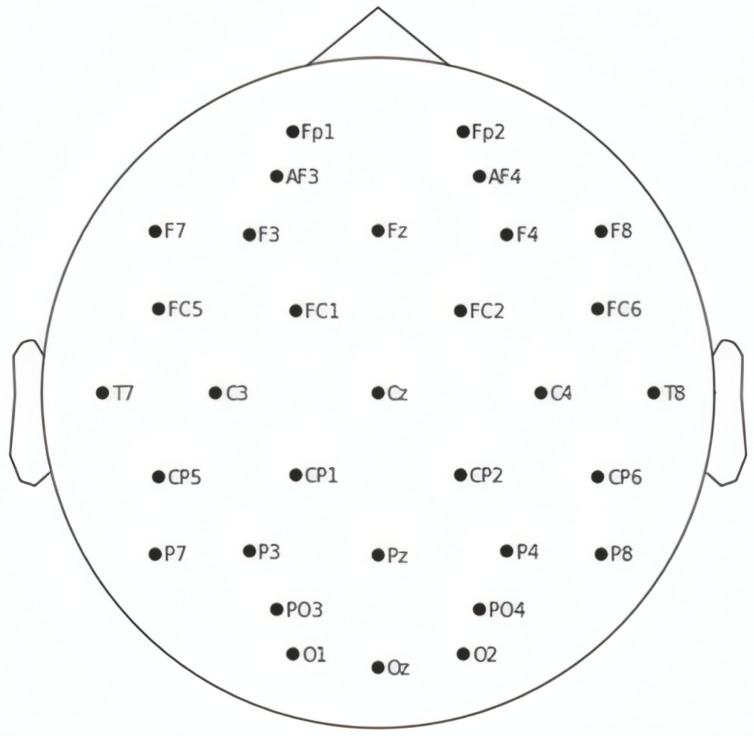
Electrode placement according to the International 10–20 system [17].

**Figure 2 sensors-25-01827-f002:**
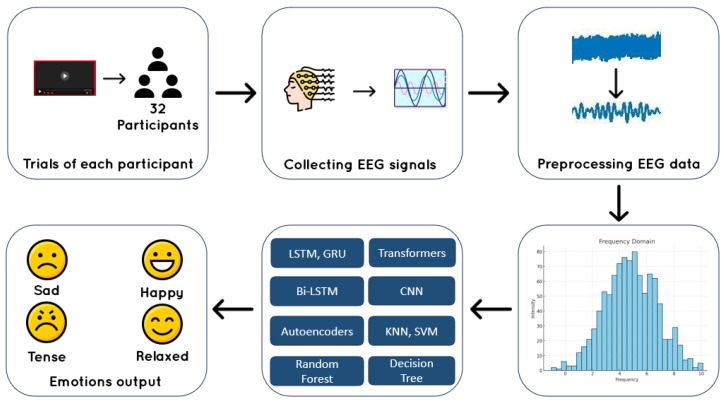
Preprocessing pipeline.

**Figure 3 sensors-25-01827-f003:**
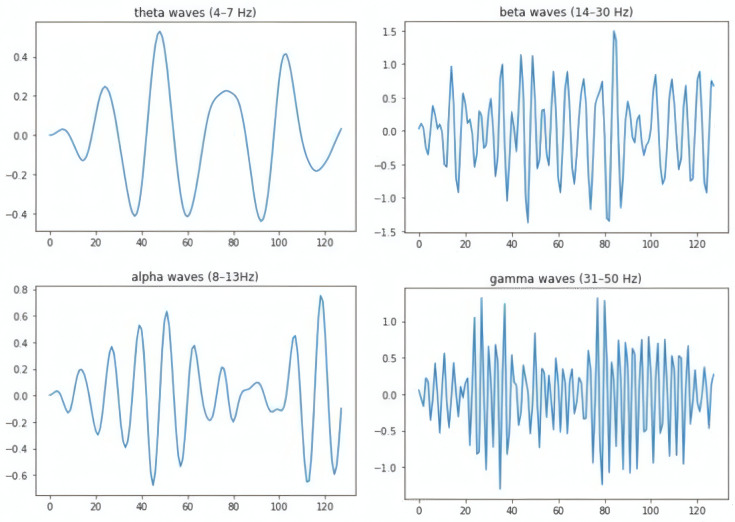
EEG frequency bands.

**Figure 4 sensors-25-01827-f004:**
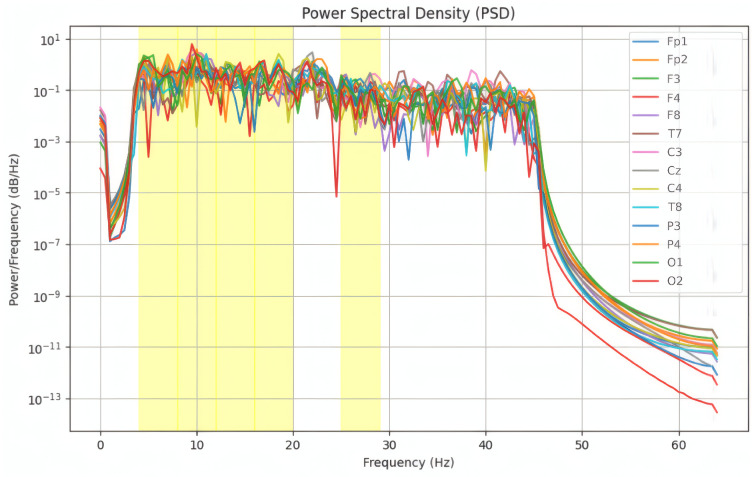
Power Spectral Density (PSD) computed via Welch’s method.

**Figure 5 sensors-25-01827-f005:**
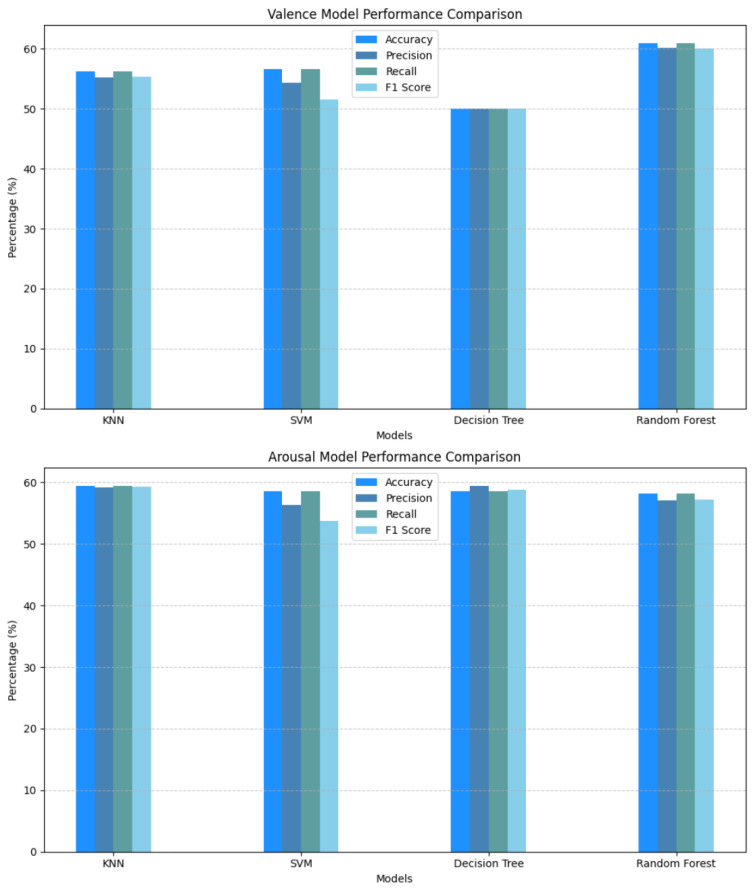
Performance comparison of classical ML models for valence and arousal.

**Figure 6 sensors-25-01827-f006:**
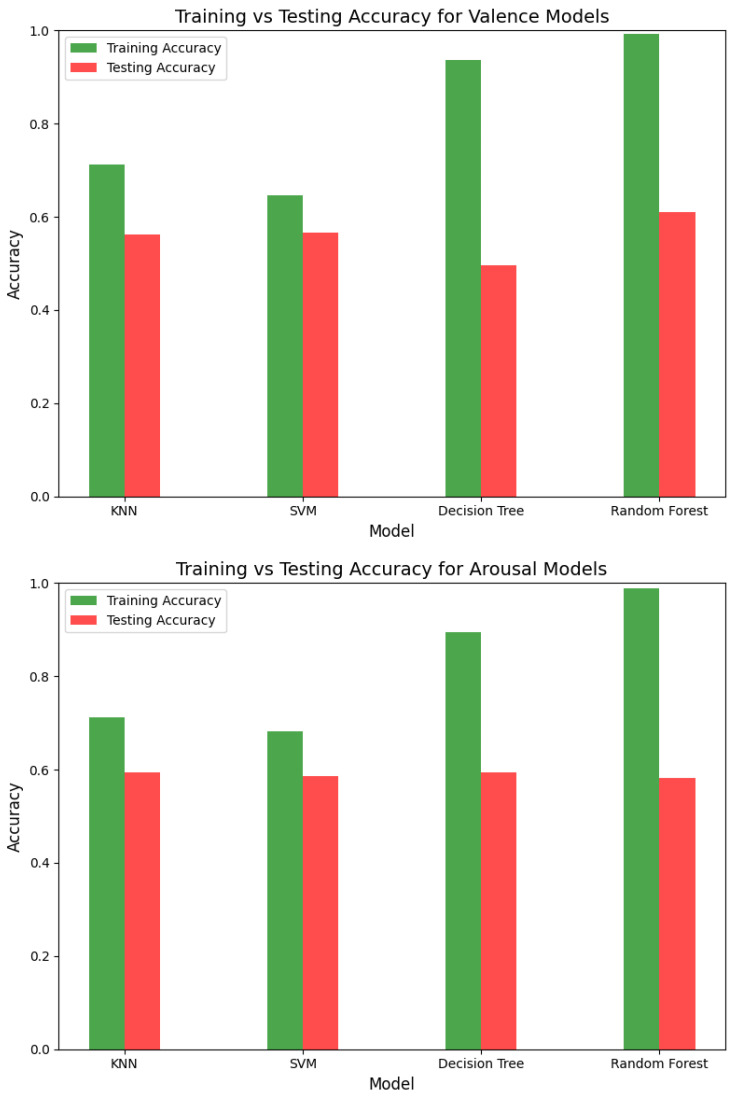
Training vs. testing accuracies for valence and arousal models.

**Figure 7 sensors-25-01827-f007:**
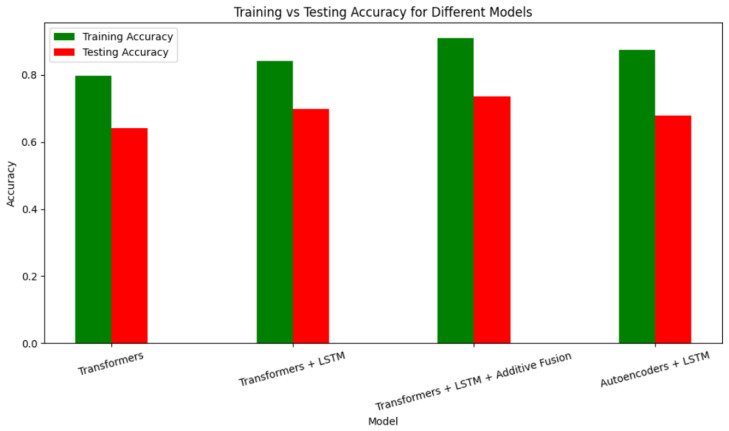
Training vs. testing accuracies for autoencoder-enabled models.

**Figure 8 sensors-25-01827-f008:**
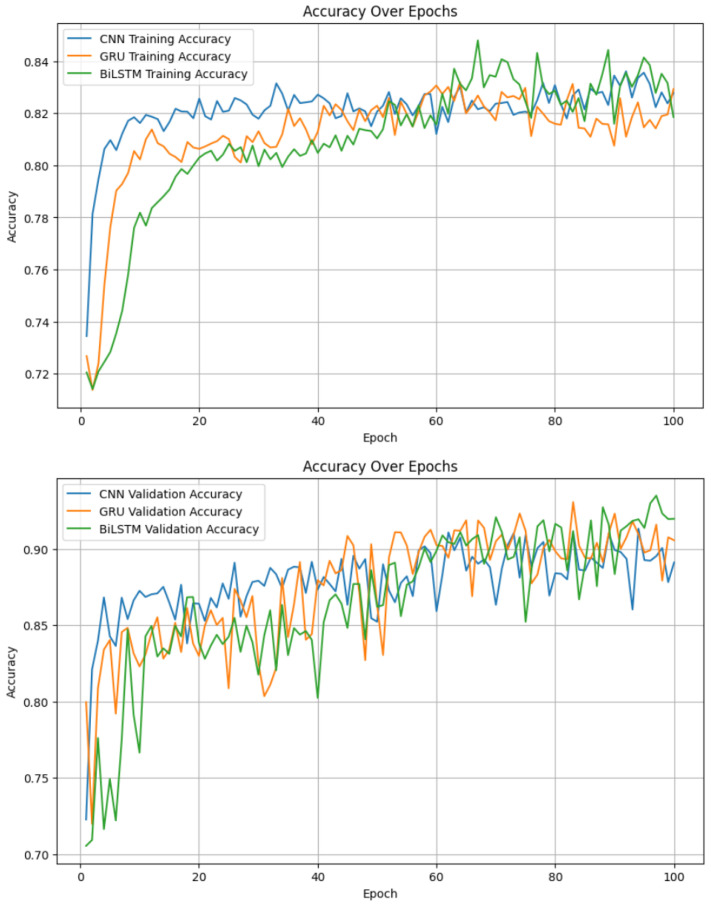
Training and validation accuracies for RNN-based models.

**Figure 9 sensors-25-01827-f009:**
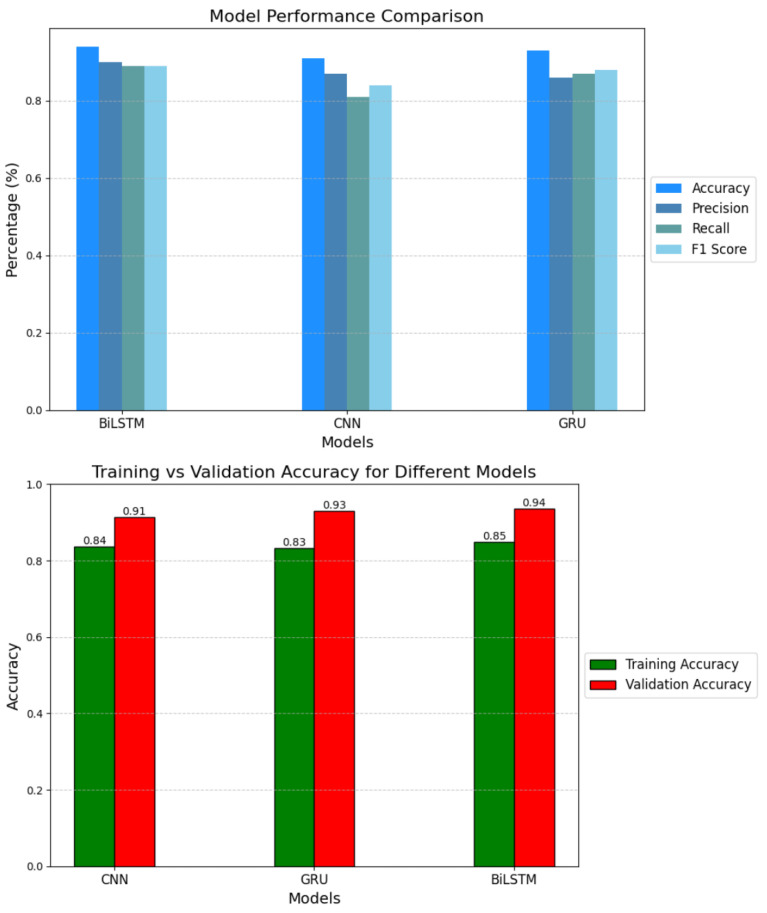
Comparison of precision, recall, F1-score, and accuracies across models.

**Figure 10 sensors-25-01827-f010:**
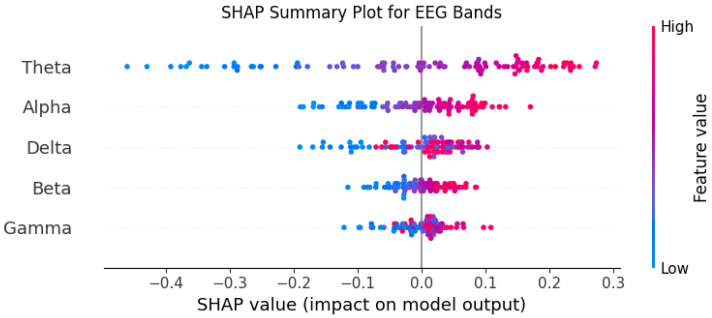
SHAP plot for the BiLSTM model, highlighting feature importance.

**Table 1 sensors-25-01827-t001:** Summary of related works on EEG-based emotion recognition using DEAP dataset.

Study	Objective	Techniques Used	Results (Accuracy)
Chen et al. [10]	Valence, Arousal Classification	CNN-based Feature Extraction	85.57% (Arousal), 88.76% (Valence)
Alhagry et al. [11]	Temporal EEG Analysis	LSTM with Cross-Validation	85.65% (Valence), 85.45% (Arousal)
Li et al. [12]	Hybrid Model for Emotion Recognition	CNN + LSTM	75.21% (Four-class classification)
Xing et al. [13]	Subject-Independent Recognition	SAE + LSTM	81.10% (Valence), 74.38% (Arousal)
Pichandi et al. [14]	Feature Extraction with SVM	AlexNet + DenseNet + PCA + SVM	95.54% (Valence), 97.26% (Arousal)
Chen et al. [15]	Attention-based Model	Hierarchical Bidirectional GRU with Attention	69.3% (0.5 s EEG segments)

**Table 2 sensors-25-01827-t002:** Accuracy comparison of classical ML models for valence and arousal.

Models	Valence		Arousal	
	Training Accuracy	Testing Accuracy	Training Accuracy	Testing Accuracy
KNN	71.3%	56.2%	71.2%	59.4%
SVM	64.6%	56.6%	68.3%	58.6%
Decision Tree	93.7%	49.6%	89.6%	59.4%
Random Forest	99.3%	60.9%	98.9%	58.2%

**Table 3 sensors-25-01827-t003:** Accuracy comparison of hybrid models.

Models	Training Accuracy	Testing Accuracy
Transformers	79.85%	64.15%
Transformers + LSTM	84.26%	69.81%
Transformers + LSTM + additive fusion	90.98%	73.58%
LSTM + Autoencoders	87.56%	67.88%

**Table 4 sensors-25-01827-t004:** Accuracy comparison of RNN-based models.

Models	Training Accuracy	Testing Accuracy
BiLSTM	85%	94%
GRU	83%	93%
CNN	84%	91%

## Data Availability

The data presented in this study are available at https://www.eecs.qmul.ac.uk/mmv/datasets/deap/ (accessed on 10 February 2024).

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
