# Peer review of "Enhancing EEG-Based Emotion Detection with Hybrid Models: Insights from DEAP Dataset Applications"

_sensors, 2025, doi:10.3390/s25061827_

Round 1

Reviewer 1 Report

Comments and Suggestions for Authors

The paper presents a pipeline for emotion detection based on EEG signals and various approaches based on multiple machine learning techniques.

Q1:

The best approach presents an average accuracy of about 85%/95%, it seems lower than other approaches presented in literature review so it is not clear why any of the presented approaches could be better than other works. A direct comparison between the results of the current work and the results presented in literature review should be presented in the Results and Discussion section. The authors should present why the presented approach is meaningful for the field.

Q2:

The abstract also mentions an application for real time emotion detection based on EEG signals but this is not thoroughly presented in the paper. It is mentioned in from line 388 to 402, but as it is mentioned as a key point, it should be more thoroughly presented or simply dismissed and mentioned as future work.

Q3:

The structure of the paper is confusing. It has the following sections:

1.         Introduction

2.         Related works

3.         Methods and Materials

4.         Methodology

5.         Results and Discussion

6.         Conclusion and Future Work

There is an overlap between the contents for sections 3, 4 and 5.

The content for section 3 is appropriate, concerning section 3.1 and 3.3. Section 3.2 seems more appropriate for the Methodology section. If the authors prefer that section 3.2 is in that section, than probably section 4, Methodology should be renamed in something more appropriate related to algorithm design or classification approaches. 

Q4:

Also, it is mentioned that the best approach uses data from only 6 subjects from the dataset. It is unclear why (the authors mentioned that using more made it challenging for the model), but it is not clear if the built model was also tested on data from only 6 subjects. In this case, is this model capable of generalizing on new data? Are the built models personalized for each subject? Were any testing performed on the data from the rest of the subjects?

Q5:

Only 4 figures are referenced in the text. The rest are not explained in the text. The authors should reference the figures in the text to better explain their relevance.

Q6:

Figure 2 is not referenced in text and no source is given, not even as a starting point. It should also be linked to the targeted classes for the built ML and DL models.

Q7:

There are yellow markings on Figure 4 (b). The yellow markings are not explained. There are two figures at number 4 and they are not separately numbered. 

Q8:

Figure 5 presents the pipeline for classifying the EEG signals, but it is too superficial and it is not referenced or described elsewhere in the text.

Also, Figure 5 presents in the last block the detection of 4 emotions. Are these the targeted emotions for all of the three implementations (the three implementations as they are presented in section 4. Methodology). Section 4 mentions that the first implementation deals with a binary classification task, the second implementation does not mention the targeted classes and the third one mentions again a binary classification problem with thresholding. If thresholding is applied in all cases, that it should be left only in section 3 (lines 122 -133) and detailed. The combinations of low/high arousal and low/high valence would result in 4 classes. These are not presented anywhere in the paper. Probably confusion matrices would help in clearing things up about the targeted classes for each approach. 

Q9:

Also, it may be beneficial to show the dimensions of the training/ testing sets in each approach.

Q10:

Table 3 presents the best results of the work. It shows that Testing accuracies are much higher than training accuracies, which is a signal that something is wrong with the data: maybe with the split of the data in training and testing subsets. Can the authors explain why testing accuracy is around 10% bigger than training accuracy in Table 3?

Minor problems:

Q:

There are many unnecessary capitalizations (abstract, line 23, caption of Figure 1, line 107, line 157, line 208 and many more)

Q:

Equations are not numbered, so they are not referenced in text.

Q:

Citations are referenced in an unclear way.

Q:

In the Related work section, it is not clear if the current work presents the model or if the referenced work. For example, at line 98: 

"In addition, this study [14] introduced a model combining… "

Is it about the study presented in [14], or the current work, based on the work presented in [14]?

The same for reference [15].

Q:

In the rest of the paper, for some citations it is not clear what statements they are backing up. For example, I don’tunderstand why source 25 is referenced at the end of that paragraph. Did [25] present the same techniques?

These techniques form the foundation  of our emotion recognition pipeline, allowing us to build robust models that can accurately classify emotional states based on brain activity patterns[25]”

Q:

Data Availability statement should be filled.

The reference section should be reviewed to comply with the guideline for this section. Some examples:

Q:

Many references with multiple authors have to many “and” between names. “and’ should be used only between the last two listed authors, and a comma should be used for separating the first ones. Reference 9 has too many authors listed.

Q:

Reference no. 23, 24 are not correctly cited, there are some unnecessary keywords or topics: Dwivedi, Amit Kumar and Verma, Om Prakash and Taran, Sachin, 2024 11th International Conference on Signal Pro-cessing and Integrated Networks (SPIN), EEG-Based Emotion Recognition Using Optimized Deep-Learning Techniques,2024, , , 372-377, Deep learning;Time-frequency analysis;Emotion recognition;Computational modeling;Brain model-ing;Feature extraction;Electroencephalography;EEG;Emotion Recognition;Deep-Learning Methods;Bayesian Optimization, 10.1109/SPIN60856.2024.10512074

Q:

26 lacks a capital for the Publication name.

Q:

28 is not the correct way to reference online resources.

Comments on the Quality of English Language

There are many sentences that need rephrasing, as in the current form they are unclear. Also, authors should review spelling. For example: 

"For the third implementation of some deep learning architectures, we have implemented severals ones to see what’s the best deep learning architecture that will give us the best results, however each one of these archirectures is different from the other one, we have implemented a GRU model, CNN model, BiLSTM model”

Author Response

We sincerely thank you for your thoughtful and constructive feedback on our manuscript. We
have carefully addressed all the comments and made the necessary revisions to improve the quality of
the paper. Below, we provide a point-by-point response to your comments. Changes in the
manuscript are highlighted in blue for easy reference.

Comment 1: Clarity and structure of the manuscript.
Response: We acknowledge that some overlap exists between the Materials and Methods section and
the Methodology section. To eliminate confusion, they were combined into one section, called Research
Design and Methodology. This restructuring will offer a clear presentation of our methods.
[Line 129]
Comment 2: Comparison with the state of the art
Response: Direct comparisons of our results have been added in the Results and Discussion section.
Besides, we also discuss why our work is still of relevance despite possible differences in accuracy: high
computational efficiency, interpretability by SHAP, and feasibility for real-time estimation.
[Line 344]
Comment 3: Presentation of the real-time application
Response: The real-time emotion detection system is presently in the simulation stage for feasibility
testing. Since the full implementation is still underway, we clarify now in the manuscript that real-time
deployment is future work. This comment has been included in both the Abstract and Conclusion&
Future Work.
[Line 33, Line 595]
Comment 4: Justification for using 6 participants
Response: It has been clarified that six participants were chosen for hyperparameter fine-tuning and for
verifying the validity of the model before courageous scaling. Moreover, our best models were validated
on the full dataset (32 participants) to ensure generalizability. This was explicitly discussed in section
3.7.
[Line 351]
Comment 5: Figures and tables
Response: To clarify, we have assured that all figures contained in this manuscript are appropriately
cited in the text. In this respect, we have specifically mentioned Figure 2 and cited the source. More
information is provided for Figure 4(b), and further revisions are carried out in Figure 5 to underline its
importance for the classification pipeline.
[Line 181, Line 208, Line 430]
Comment 6: Clarity of results and metrics
Response: The key evaluation metrics such as F1-score, Recall, Specificity, and Sensitivity are reflected
1
by Figures 5 and 9. Importantly, we have provided in Section 4.4 an explanation for the 10% gain in
test performance, a result of regularization techniques, particularly dropout and early stopping, as well
as proper feature selection methods.
[Line 445]
Comment 7: Language quality and references
Response: The sentences have been revised to enhance readability and rectify grammatical errors.
Citations attach themselves to the BibTeX citation style following standard academic formatting. The
references have been directly pulled from the original sources leaving out any manual alteration. We,
however, made sure to keep consistency in the citation formatting in the whole manuscript.
[Line 636]

Reviewer 2 Report

Comments and Suggestions for Authors

The manuscript explores the application of various machine learning and deep learning techniques for emotion recognition using EEG signals. The results from the DEAP dataset demonstrate the effectiveness of neural architectures in recognizing emotional states from EEG signals, achieving up to 90% accuracy. Furthermore, a real-time emotion detection system is developed. The objective of this work is clear, and the proposed technique sounds available. However, lots of vital issues must be improved, as listed below:

1. The manuscript contains many acronyms. Please include a table that clarifies the acronyms employed, provides their corresponding full names, and describes their usage.

2. Please highlight the motivation and contribution of this work at the end of the Introduction.

3. In Section 2, the authors should also introduce the five brain rhythms and then concentrate on the related works in this field.  

4. When discussing previous studies, it is better to compare the methods used with those employed in your work. This will help readers understand the improvements in your approach. Please summarize them into a table.

5. To facilitate reproducible research, I suggest that the authors release the programming codes of the proposed method and then provide the link in the "Data Availability Statement" of the manuscript. It would have a positive effect on the academic community.

6. What is the number of samples for different emotional states in the DEAP dataset? How do you deal with the issue of data imbalance? Why is the threshold at 6.5, not 5?

7. Using data from only six participants for the deep learning models is unjustified. Please provide a rationale for discussing any potential limitations it may introduce.

8. Regarding the figures, their fonts are too small and have a low resolution. Please improve them.

9. The paper's strong point is its application of SHAP. I suggest expanding the discussion to include how these insights could further refine the models and the feature selection process.

10. What are the parameters of SVM? How do you set the k value in kNN?

11. More evaluation metrics, such as F1-score, recall, specificity, and sensitivity, are required.

12. The manuscript emphasizes real-time detection, so what time is required for the proposed method? Is it faster than those existing methods? What about the time complexity? How do you prove that this method can better satisfy the requirements of real-time detection?

13. Last but most importantly, please clarify the novelty and advantages of the proposed methods compared to the existing works. Currently, the manuscript can not provide deep insights into this field. After reading it, I see that the whole contribution seems weak, like repeated work.

Comments on the Quality of English Language

The authors should spend time revising this manuscript. The flow of this work needs to be improved. I advise that the authors review and follow those high-quality papers in this field.

Author Response

We sincerely thank you for your thoughtful and constructive feedback on our manuscript. We
have carefully addressed all the comments and made the necessary revisions to improve the quality of
the paper. Below, we provide a point-by-point response to your comments. Changes in the
manuscript are highlighted in blue for easy reference.

Comment 1: Motivation and contribution.
Response: The Introduction presents clear motivation toward the highly interpretable, real-time EEG-
based emotion recognition system. Also, contributions are at last outlined at the end of the Introduction.
[Line 87]
Comment 2: Related work and bibliography.
Response: We have inserted a comparative table in the manuscript, which presents prior studies in
datasets, models, and accuracies. Moreover, a comprehensive analysis comparing previous methods to
ours, emphasizing salient differences and strengths of our proposed model, has been incorporated.
[Line 128]
Comment 3: Methodological details.
Response:In Section 3, we presented the five frequency bands of EEG so that the signal processing
framework will become clear. Again, we clarified the threshold selection process by stating that 6.5 was
chosen instead of 5 based on prior studies for a stronger class separation, and hence a more balanced
dataset distribution. In Section 3.1 we have included an acronym table for clarity. Also, our code is
publicly available, and a link to it has been added to the first paragraph of the Abstract, to make it
replicable. Regarding some model parameters, we would like to say that the SVM classifier uses the RBF
kernel with C=1.0, while for KNN, we used k=7 selected by cross-validation.
[Line 145, Line 130, Line 18]
Comment 4: Justification for the reduced sample size.
Response: We used a small group of six subjects for hyperparameter tuning and initial evaluation.
This was to help set the method straight, optimize compute resources, and test for model feasibility
prior to scale. The final evaluation was performed with all 32 participants in the dataset to provide
robust generalizability to these findings. This complete validation would ensure that the model does not
suffer from a bias against a smaller sample used for initial tuning, thus guaranteeing its performance
across a wide variety of individuals.
[Line 351]
Comment 5: Explanation of results and SHAP.
Response: The more considerable section about SHAP interpretation describes how feature importance
insights assist in optimizing model performance and feature selection. We shall compare real-time per-
formance against previous methods and outline the analysis of our computational complexity in Section
4.8.
[Line 493]
Comment 6: Visual and linguistic improvements.
Response: We have reformulated every single figure in a higher resolution to increase clarity and make
them easier to understand by the general audience. We have also worked on the text in the manuscript
to make it clearer, refine sentence structure, and enhance the flow of ideas in the manuscript

Reviewer 3 Report

Comments and Suggestions for Authors

Dear authors,

Upon reviewing the manuscript titled "Enhancing EEG-Based Emotion Detection with Hybrid Models: Insights from DEAP Dataset Applications," I would like to provide the following feedback concerning its submission.

The manuscript is rather concise, particularly given the intricate nature of the subject matter. Emotion recognition is a complex field of study, encompassing various ethical and privacy concerns that the authors have not adequately addressed. It is imperative for the authors to engage in a more comprehensive examination of these issues, as well as the related implications, to enhance the overall quality of the manuscript and to offer more pertinent insights for researches interested in this area of investigation.

Furthermore, I encountered difficulties while reading the submitted paper due to its lack of clarity. The contributions towards the improvement of EEG-based emotion detection are not distinctly articulated, and the organization of the content does not align with the expected standards for publications in the Sensors Journal.

Here are several specific recommendations to enhance the manuscript for future submissions:

**Introduction**  

1) The current introduction presents a somewhat superficial view of the approaches to the issue of emotion recognition in artificial intelligence. To enrich this section, including more nuanced examples showcasing the applications and implications of emotion recognition techniques would be beneficial. Furthermore, it is pertinent to address the regulatory landscape, particularly in Europe, which has legislated restrictions on automating emotion recognition technologies. This raises critical ethical and regulatory considerations that warrant thorough examination in the introduction.

2) Additionally, it is important to highlight the contributions of this work more effectively. Strengthening this aspect will provide clearer insights into the value added by your research.

3) Toward the conclusion of the introduction, please include a paragraph outlining the structure of the paper. A brief summary of the following sections will guide the reader and clarify the content's organization.

4) Lastly, pay attention to citation formatting throughout the manuscript. Specifically, ensure a single space before the opening bracket in all citations. For example, revise “experience[1].” to “experience [1].” This adjustment should be consistently applied to all references within the text.

** 2. Related Works **

1) This section is crucial for comparing your results with those published in the scientific literature that addresses similar topics. However, the connection between the selected paper and your work is not clearly articulated in the text. Additionally, there is no explanation regarding the selection process of the papers or how the databases were searched, which is important for establishing the credibility of your research. The criteria used for paper selection should be made explicit to the readers. Therefore, I recommend including this information and assessing whether this paper is significant for your theme, along with a rationale for its relevance.

2) Did all these papers utilize machine learning applications from the DEAP database? I recommend creating a table summarizing the main related studies, their objectives, the techniques employed, and the results. This approach could help compare your results with the related works and provide insights into your work.

3)The source for Figures 1 and 2 is currently missing.

4) In Section 3.3, I recommend explaining more about the AI tools and techniques used and their relation to this work.

**3. Methods and Materials; 4. Methodology and more**

1) I recommend a thorough reevaluation of this section for improved clarity. The current structure is disorganized and may lead to reader confusion. Merging the two topics into a single, well-structured section would enhance coherence and detail. The existing separation into "Methods and Materials" and "Methodology" creates redundancy and obscures understanding.

2) The rest of the manuscript focuses on disorganization; the authors should explicitly state why they chose different techniques and what kind of results can be expected. This is not clear throughout the manuscript.

3) The concluding remarks are insufficient, as they fail to encapsulate the objectives of the work and the key findings effectively.

Considering these concerns, I have decided to reject the submission. I hope this review helps you enhance your work.

Best regards,

Author Response

We sincerely thank you for your thoughtful and constructive feedback on our manuscript. We
have carefully addressed all the comments and made the necessary revisions to improve the quality of
the paper. Below, we provide a point-by-point response to your comments. Changes in the
manuscript are highlighted in blue for easy reference.

Comment 1: Introduction
Response: An enhancement of the introduction with concrete applications of EEG-based emotion
detection is presented. In this regard, the relevance of this technology in the mental health monitoring,
human-computer interaction, and adaptive learning systems context has been elaborated in an effective
manner. In addition, the introduction has been supplemented with a discussion of European regulations,
particularly the General Data Protection Regulation (GDPR) restrictions on biometric data, to touch on
ethical and legal issues. The contributions of the study are clearly outlined now, underscoring uniqueness
and significance of the paper. At last, a section roadmap has been added at the end of the introduction,
guiding the reader at this moment through the structure of the paper.
Line 48, Line 108
Comment 2: Related Works
Response: Section 2 discusses criteria for the selection of related studies for a clear outline of how
relevant research studies were selected. Besides a comparative table to show if the studies used the
DEAP dataset, this also helps readers with a much clearer understanding of the depth of the analysis
done. Further, citations have been added to figures now shown, which ensures proper attribution and
improves the credibility of the figure.
Line 109 - 128
Comment 3: Reorganization of the methodology
Response: The Methods and Materials and Methodology sections have been merged into ”Research
Design and Methodology” to remove redundancy.
Line 129
Comment 4: Presentation of results
Response: The Results section was reorganized for greater clarity, ease of reading, and better presen-
tation of key findings. More explanations were provided to justify specific conclusions from the data so
that a clearer connection could be drawn between the results of this study and the objectives of research.
This reorganization additionally improves the flow of the entire section, creating a tighter bond between
findings and their importance.
Line 421 - 549
Comment 5: Conclusion
Response: The Conclusion has been revised to better summarize key findings and outline future research
directions.
Line 550 - 635

Round 2

Reviewer 1 Report

Comments and Suggestions for Authors

The authors responded to all the suggestions. The new version is much more clear and more logically structured.

There is one more aspect to consider: the equations should be numbered and mentioned in text.

Author Response

Comment: The authors responded to all the suggestions. The new version is much more clear and
more logically structured. There is one more aspect to consider: the equations should be numbered and
mentioned in text.

Response: We have now ensured that all equations are numbered and appropriately referenced in the
text. Each equation is introduced within the relevant section and cited using its corresponding number
to enhance clarity and logical flow.
Line 197 - Line 204 - Line 317 - Line 327 - Line 376

Reviewer 2 Report

Comments and Suggestions for Authors

The authors have addressed the concerns raised in my previous queries, and the revisions have enhanced the overall quality of the manuscript. My final comment is to improve the quality of the figures. First, for Figure 4, please provide the name of the channels, such as FP1, FP2, and so on, not channel 1, channel 2. Second, the fonts in most of the figures are too small, which makes the details very difficult to read. Please check and improve them.

Author Response

Comment: The authors have addressed the concerns raised in my previous queries, and the revisions
have enhanced the overall quality of the manuscript. My final comment is to improve the quality of the
figures. First, for Figure 4, please provide the name of the channels, such as FP1, FP2, and so on, not
channel 1, channel 2. Second, the fonts in most of the figures are too small, which makes the details very
difficult to read. Please check and improve them.

Response: We updated the figure to include all the correct channel names (e.g., FP1, FP2, etc.) for
improved clarity. Additionally, we reviewed all the figures, increasing the font sizes to enhance legibility
and overall readability.
Line 200

Reviewer 3 Report

Comments and Suggestions for Authors

Dear authors,

The manuscript has been improved, but I would like to emphasize one thing about Subsection 3.1 Acronymous.In MDPI format, please see Section 8-Back Matter for recommendations on including supplementary material.

The Instructions for authors can be found at the following link:

https://www.mdpi.com/authors/layout#_bookmark91

Sincerely,

Author Response

Comment: The manuscript has been improved, but I would like to emphasize one thing about Sub-
section 3.1 Acronymous.In MDPI format, please see Section 8-Back Matter for recommendations on
including supplementary material. The Instructions for authors can be found at the following link:
https://www.mdpi.com/authors/layout#_bookmark91
Response: Regarding your comment on Subsection 3.1 (Acronyms) and the MDPI formatting require-
ments for supplementary material, we have now moved the list of acronyms to the Back Matter section
using the appropriate MDPI formatting. Specifically, we have included the acronyms under the ’Abbre-
viations’ section, as per the MDPI guidelines (Section 8.9).
Line 625 - 628